# Genomic Interplay between Neoneurogenesis and Neoangiogenesis in Carcinogenesis: Therapeutic Interventions

**DOI:** 10.3390/cancers15061805

**Published:** 2023-03-16

**Authors:** Zodwa Dlamini, Richard Khanyile, Thulo Molefi, Botle Precious Damane, David Owen Bates, Rodney Hull

**Affiliations:** 1SAMRC Precision Oncology Research Unit (PORU), DSI/NRF SARChI Chair in Precision Oncology and Cancer Prevention (POCP), Pan African Cancer Research Institute (PACRI), University of Pretoria, Pretoria 0028, South Africa; 2Department of Medical Oncology, Faculty of Health Sciences, Steve Biko Academic Hospital, University of Pretoria, Pretoria 0028, South Africa; 3Department of Surgery, Steve Biko Academic Hospital, University of Pretoria, Pretoria 0028, South Africa; 4Centre for Cancer Sciences, Division of Cancer and Stem Cells, Biodiscovery Institute, University of Nottingham, Nottingham NG7 2RD, UK

**Keywords:** neural stem cells, neurotransmitter, neurotrophin, growth factor, axonogenesis, metastasis, tissue remodeling, reactive oxygen species, neurogenesis, angiogenesis

## Abstract

**Simple Summary:**

This review describes how the two processes of angiogenesis (the generation of new blood vessels) and neurogenesis (the generation of new nerve fibers) act together to drive cancer progression. It also describes how they are both associated with a lower rate of patient survival. These two processes share signaling pathways and, in many cases, the initiation of one leads to the initiation of the other. Both processes require tissue alterations and are reliant on cell migration. They favor cancer progression by supplying the tumor with nutrients and facilitating communication/movements within the tumor. Thus, these two processes contribute to the spread of cancers, as tumors can use nerve fibers and blood vessels as a routes to migrate from the initial point of cancer development to the surrounding area or to the distal sites of the body.

**Abstract:**

Angiogenesis, the generation of new blood vessels, is one of the hallmarks of cancer. The growing tumor requires nutrients and oxygen. Recent evidence has shown that tumors release signals to attract new nerve fibers and stimulate the growth of new nerve fibers. Neurogenesis, neural extension, and axonogenesis assist in the migration of cancer cells. Cancer cells can use both blood vessels and nerve fibers as routes for cells to move along. In this way, neurogenesis and angiogenesis both contribute to cancer metastasis. As a result, tumor-induced neurogenesis joins angiogenesis and immunosuppression as aberrant processes that are exacerbated within the tumor microenvironment. The relationship between these processes contributes to cancer development and progression. The interplay between these systems is brought about by cytokines, neurotransmitters, and neuromodulators, which activate signaling pathways that are common to angiogenesis and the nervous tissue. These include the AKT signaling pathways, the MAPK pathway, and the Ras signaling pathway. These processes also both require the remodeling of tissues. The interplay of these processes in cancer provides the opportunity to develop novel therapies that can be used to target these processes.

## 1. Introduction

The hallmarks of cancer are characteristic alterations in the cells that accompany or lead to the development and progression of cancer. One of these hallmarks is uncontrolled or dysregulated angiogenesis. This allows the tumor mass to be provided with nutrients as a result of the development of new blood vessels (reviewed in [1]). Recently, evidence brought forward shows how the creation of new nerve tissue, neurogenesis, is another important contributing factor to the development and progression of cancer [2,3,4]. Multiple studies indicate that tumors in tissue with a high level of blood vessels present are more likely to develop intra-tumoral neural infiltration, a condition that is associated with a poor prognosis. The more extensive this intra-tumoral nerve density is, the more severe the metastatic potential of the tumor and the poorer the prognosis is for the patient [5,6,7]. In the body, the distribution of blood vessels and nerves mirrors each other (Figure 1A). This is due their embryological and growth factor similarity - they share many of the same molecules and signaling pathways that guide their growth [8]. These shared molecules and pathways may be the reason why the molecules that induce angiogenesis during cancer also induce neurogenesis. Examples of the interaction between the nervous tissue and the surrounding blood vessels include the blood brain barrier (BBB), where astrocytes and pericytes around blood vessels interact with these blood vessels to aid in the regulation of the BBR. Interactions between the nervous tissue and the blood vessels surrounding it are known as neurovascular units (NVUs) (Figure 1B) [9]. Brain tumors are known to alter the structure of the blood brain barrier (BBB) into what is known as the blood–tumor barrier (BTB). It is now known that cancers can drive neurogenesis, axonogenesis, and the repurposing of existing nerve fibers.

The new nervous tissue formed during neurogenesis is also dependent on new blood vessels to supply the nervous tissue with oxygen and nutrients to ensure the new tissues’ continued survival [10]. This review discusses the role played by the interplay between neurogenesis and angiogenesis in cancer, and although many of the examples discussed come from studies on the interplay of these processes following stroke, the pathways discussed are expected to be the same or similar to the pathways exploited by cancer cells to attract or create nerve fibers or increase the vascularization to allow tumors to grow and metastasize. During metastasis, cancer cells access the circulatory system, move, and implant in distant organs. This process involves reciprocal communication between cancer cells, endothelial cells of the vasculature, and neural cells of the nervous systems. This communication correlates with poor prognosis [11].

## 2. Angiogenesis and Neurogenesis

The reciprocal communication between tumors and the nervous system is evidenced by the fact that cancer patients experience cancer related pain [12] as a result of neuro-oncogenic pressure on the fibers as the tumor volume increases [13], secretion of stimulatory factors on peripheral fibers with depolarizing effects, axon demyelination, and pathological neural plasticity induced by tumor-derived factors [14].

The term angiogenesis describes the generation of new blood vessels. There are two main methods whereby angiogenesis occurs. Vasculogenic involves vascular progenitor cells forming new blood vessels. These circulating endothelial progenitor cells (EPCs) are derived from the bone marrow [15]. These cells possess surface markers, such as CD34, CD31, VEGFR-2, and Tie-2. Tumor cells are known to secrete VEGF and cytokines such as stromal-derived factor-1. These pro-angiogenic factors recruit the circulating progenitor cells and stimulate proliferation. These pro-angiogenic stimuli also activate the matrix metalloproteases to break down the extracellular matrix, allowing cells to migrate and penetrate cell layers [16].

Sprouting angiogenesis involves the formation of new vascular structures from an existing vessel. This, firstly, involves the destabilization of endothelial cells and endothelial mesenchymal transition. Once again, activated proteases are required to degrade the ECM and basement membrane, allowing for directed migration and proliferation [16]. Once the cells form the lumen and tubulogenesis of the new blood vessels, the transition is reversed, with the cells reverting to the resting state [17]. This process involves the VEGF and Notch signaling pathways [18]. As such, sprouting angiogenesis is a process whereby new blood vessels are formed from pre-existing blood vessels. This process occurs in response to both mechanical and chemical stimuli and is important for normal development and wound healing. It facilitates tumor survival and progression as it provides cancer cells with oxygen and nutrients to sustain their growth (reviewed in [19]). During angiogenesis, new micro vessels are formed as they branch off from pre-existing vessels [20].

Angiogenesis involves processes, such as proliferation of endothelial cells and the formation of tube-like vascular structures, as well as branching and anastomosis (reviewed in [21]). This review will mainly focus on sprouting angiogenesis. Both pro- and anti-angiogenic factors act to regulate angiogenesis, with any changes in the balance of these factors either promoting or inhibiting angiogenesis. Pro-angiogenic factors are released by endothelial cells, monocytes, and tumor cells. Angiogenesis also requires the activation of various processes, such as basement membrane degradation, endothelial cell proliferation, migration, and cell remodeling, to form a tube or blood vessel (reviewed in [22]). Angiogenesis can be promoted by adrenergic signaling, which is thought to contribute to the formation of preneoplastic niches and decreased immune function [23]. β-adrenergic receptor activation has been found to be essential in the progression and malignant growth of ovarian [24], pancreatic [25], and pulmonary [26] cancers.

In adult mice, neurogenesis predominantly occurs in the subventricular zone (SVZ) and the hippocampal dentate gyrus (DG), and it is thought that these two areas are where neural stem cells are found within the adult brain [27,28]. Different types of cells are involved in neurogenesis. The slow-dividing or true stem cells are mostly quiescent but activate and divide asymmetrically to self-renew and give rise to immediate progenitor cells [27]. These intermediate progenitor cells divide rapidly to give rise to nervous tissue. In addition to these progenitor cells, other types of cells integrate into existing neuronal networks. These include neuroblasts [29,30] and newborn neurons, which are electro-physiologically active [31]. Neurogenesis can be activated or inhibited by a multitude of signals, including growth factors, cytokines, chemokines, neurotrophins, steroids and extracellular matrix components, the activation of specific transcription factors, and signal transduction pathways (reviewed by [32]). Additional signals may result from environmental stimuli, such as exercise alterations in an organism’s environment, stress, or social isolation [33,34]. The growth and migration of tumors around and along nerve fibers are known as perineural invasion [35]. Additionally, it is now known that nerves actively grow into and throughout cancer tissue, resulting in increased metastasis as these nerves provide signals to the cancer as well as a pathway to migrate along [36]. Autonomic neurotransmitter receptors can stimulate cancer cell growth through the activation of corresponding signaling pathways [37]. In its most basic form, the crosstalk between neurogenesis, angiogenesis, and carcinogenesis is based upon common signaling pathways and chemokines. Cancer cells express neurotrophic and pro-angiogenic markers. In many cases, these molecules are both neurotrophic and pro-angiogenic. The infiltration of new nervous tissue into cancer cells gives cancer cells a route to migrate along in the same way that these cells can use lymphatic and blood vessels. In other words, the signaling pathways activated by more cells stimulate both neurogenesis and angiogenesis while assisting in metastasis [38].

## 3. Stem Cell Niches and Blood Vessels

In adults, the stem cells that give rise to new tissue are in microenvironments known as stem cell niches. It is the communication between the stem cells and these niches which governs the activity of these stem cells, instructing them when to proliferate and differentiate. This also involves niche cells regulating stem cell activity in response to external signals.

This close association between blood vessels and stem cells has been observed for hematopoietic stem cells [39] and neuronal stem cells [40,41,42]. When it comes to neurogenesis, neural stem cells, otherwise known as neural progenitor cells, migrate through blood vessels to the tumor site from the neurogenic regions of the brain and metastatic niches. These cells express the doublecortin (DCX) surface marker for progenitor cells, which has been found to be expressed at higher levels in individuals with high-risk cancers. Once these cells reach the tumor site, they differentiate into various noradrenergic, mature neuronal cells [43]. Cancer stem cells can also form new neurons that are themselves capable of stimulating tumor growth in xenograft models [44]. These neural stem cells are surrounded by an extracellular matrix, rich in both blood vessels and laminin. Neurogenesis can also be induced or inhibited through direct physical contact and, therefore, signaling via integrins between the blood vessels and neural stem cells. One of these signals is the laminin receptor for α6β1integrin, which is expressed by both the stem cells and blood vessels [41,45]. Following the inhibition of α6β1integrin, the levels of both stem cell proliferation and migration away from blood vessels increase [41]. Blood vessels nearby and towards the SVZ vascular niche can interact with the SVZ, leading to the release of angiogenic factors from the surrounding endothelial cells, resulting in neurogenesis [41]. One of these factors is the chemokine CXCL12 (SDF1), which initiates a signaling pathway, resulting in the migration of neural stem cells to blood vessels and surrounding endothelial cells that express CXCL12 [46]. Endothelial cells are also able to secrete factors that inhibit neurogenesis. These include sphingosine-1-phosphate and prostaglandin-D2, both of which are G-protein-coupled receptors (GPCRs), ligands that keep neural stem cells in a quiescent state [47]. Another factor secreted from endothelial cells is neurotrophin-3 (NT-3), which is required to sustain and regulate neurogenesis [48].

## 4. Ischemic Stroke Models Providing Evidence for the Interaction of Neurogenesis and Angiogenesis

The first evidence showing the association between neurogenesis and angiogenesis was from stroke models. In these models, it was noted that recovery from the stroke resulted in increased numbers of neural stem cells and increased levels of angiogenesis [49,50]. Many of these models demonstrated the influence of therapies to treat stroke victims on the processes of neurogenesis and angiogenesis, for example, the use of stem cell therapy. Rat models of stroke showed that mesenchymal stem cell (MSC) therapy is due to factors secreted by these stem cells rather than these cells replacing lost tissue. The effect of these secreted factors rather than the cells was tested using a Wistar rat model of stroke where the rats underwent cerebral artery occlusion. These secreted factors were able to increase the levels of neurogenesis and angiogenesis in these rats, which was associated with a decrease in the neurodegenerative effects following stroke [51]. Surgical interventions for the treatment of stroke, namely, deep brain stimulation (DBS), which involves the electrical stimulation of nerve tissue, were performed. Rat models of ischemic stroke showed that recovery from stroke following DBS treatment was associated with increased proliferation and movement of cells from the subventricular zone. This is accompanied by an increase in the expression of growth factors and the stimulation of endogenous neurogenesis and angiogenesis [52]. The migration of neural progenitor cells has been shown in these models to be closely associated with blood vessels, with these vessels guiding the movement of these cells [53]. The expression of genes, miRNAs, and lncRNAs associated with these processes and recovery following ischemic stroke has shown that there are many complex signaling processes involved in neurogenesis, angiogenesis, and final recovery (reviewed in [10]).

The use of these rat models identified many important molecules involved in angiogenesis, neurogenesis, and cell movement. These include miRNA-9, which links neurogenesis and angiogenesis through regulating expression of VEGF-A by targeting transcription factors that induce VEGF-A expression [54]. Mouse models demonstrated that miRNA-126 promotes proliferation, migration, angiogenesis, and neurogenesis after brain ischemia by inhibiting its target tyrosine–protein phosphatase non-receptor type 9 (PTPN9) and activating AKT and ERK signaling pathways [55]. Signaling pathways that were identified to play a role in these processes include STAT3 pathways [56], sonic hedgehog [57], and EphA receptor-mediated signaling [58].

## 5. The Remodeling of Tissues and the Proliferation and Migration of Cells during Angiogenesis and Neurogenesis

When malignant cells invade surrounding tissues, they displace the normal cells or integrate them, altering their function. These cells that are co-opted by cancer cells to contribute to the survival of the tumor include fibroblasts [59], endothelial cells [60], immune cells [61,62], and neuronal extensions [63]. Metastatic tumors then need to form capillaries to supply oxygen and nutrients but also to act as routes for further cell movement for metastatic dissemination [64]. Nerve fibers can provide similar functions by providing the cancer cells with nerve cell signals as well as serving as migrations routes [65]. Both nerve cells and the endothelial cells that become blood vessels are attracted to and co-opted by the cancer cells [66]. Neurotrophins, such as nerve growth factor (NGF), can be released by leukocytes, such as macrophages and mast cells, to promote an axonogenic switch resulting in tumor innervation. Many immune cells express NGF [67] following induction by IL-1β [68]. This can occur during inflammatory pain and neurogenesis. In a mouse model of arthritis, the activation of macrophages results in increased levels of innervation, related to and in conjunction with angiogenesis [69].

Neurogenesis and angiogenesis require increased proliferation of neural and endothelial cells. This can be achieved by altering the length of the cell cycle, shortening it, and allowing for more cycles of division [70]. Alternatively, the rate of different types of divisions can be altered since symmetric divisions will increase the number of stem cells, while asymmetric division will result in a differentiated cell and one stem cell being generated [71]. 

The beta-adrenergic receptors, β2/3 receptors, have been shown to be involved in tumor development and progression. In a mice model, the lack of the β2 or β3 receptor led to a delay in tumor growth and angiogenesis [5,72]. In prostate cancer, these receptors lead to the stimulation of endothelial cells, resulting in angiogenesis by metabolic adjustments. The sympathetic nerves release noradrenaline, which activates β2-signaling in endothelial cells, leading to the expression of the mitochondrial cytochrome c oxidase component, COA6. This results in a decrease in normal oxidative respiration and the induction of angiogenesis [72]. This noradrenaline also promotes vascular endothelial growth factor (VEGF) expression, leading to angiogenesis [73].

### 5.1. Migration and Remodeling

Both angiogenesis and neurogenesis require the initiation or alteration of cell migration as well as the remodeling of tissue, which, themselves, require the breakdown of the extracellular membrane. In order to accomplish this altered migration and tissue remodeling, there must be specific interactions between cells, such as the immature migrating neuroblasts, astrocytic processes, and blood vessels [74]. Some of the molecules involved in these processes include stromal-cell-derived factor-1 (SDF-1), CXC chemokine receptor 4 (CXCR4), monocyte chemoattractant protein-1 (MCP-1), and matrix metalloproteinases (MMPs).

### 5.2. SDF-1 and CXCR4

The alpha chemokine SDF-1 protein is normally excreted from the ependymal cells; it is also known as CXCL12 and, under normal conditions, induces neural stem cell (NSC) quiescence [75]. However, under abnormal conditions such as in ischemic shock, SDF-1 is released from reactive astrocytes and vascular cells and, in these cases, leads to increased activation of NSCs [46]. The receptor for SDF-1, CXCR4, is also normally expressed in bone marrow, where both CXCR4 and SDF-1 are involved in hematopoietic stem cell mobilization and trafficking of NSCs [76]. In a similar way, it is thought to mobilize and direct neuroblasts. The migration of NSCs in a rat stroke model was inhibited following the blocking of SDF-1 function using an antibody against CXCR4 [77]. SDF-1 may be one of the factors released by endothelial cells that leads to the directing of migrating neuroblasts to the vasculature [46].

### 5.3. MCP-1

The (CC) family member chemokine, monocyte chemoattractant protein-1 (MCP-1), is known to interact with the CC-chemokine receptor-2 (CCR2). CCR2 is widely expressed on NSCs, and the binding of MCP-1 leads to increased NSC migration in vitro [78]. It is suspected that this is accomplished through the receptor and ligand activating the PI3 kinase pathway [79]. 

### 5.4. MMPs

Matrix metalloproteinases (MMPs) are known to have a role in cancer, promoting metastasis through the breakdown of the extracellular matrix (ECM). This then allows cells to migrate through tissue layers. This family of proteases also seems to play a role in the migration of SVZ neuroblasts. MMPs-3 and -9 are expressed in neuroblasts and their inhibition results in reduced neuroblast migration [80]. MMPs-2 and -9 act on the vasculature to assist in the migration of neuroblasts by activating the PI3K/Akt and ERK1/2 signaling pathways [77]. 

## 6. Growth Factors

Growth factors are responsible for guiding the growth of both blood vessels and nerve fibers to new target tissues or destinations. These growth factors include VEGF, NGF, and insulin growth factor (IGF). Fibroblast growth factor 2 (FGF2) stimulates neurogenesis [81], whilst bone morphogenic protein 4 (BMP4) has an inhibitory effect on neurogenesis [82]. The pituitary-gland-derived hormone, prolactin, induces neurogenesis [83], while growth differentiating factor 1 (GDF11) augments neurogenesis in older adults where the normal levels of neurogenesis have declined [28]. Brain-derived neural factor (BDNF) and pigmented epithelium-derived factor (PEDF) are both secreted from the blood vessels and endothelial tissue and both induce neurogenesis [84]. Tumor-associated neoneurogenesis involves the activity of growth factors, which generally have pleiotropic functions and serve to attract nerve fibers to a tumor and initiate neural cell growth [7]. Cancer cells also release axon guidance molecules (netrins) to aid in nerve infiltration [85]. Neurotrophic growth factor receptors and neurotransmitter receptors (NTRs) are both expressed in neurons and cancer cells. This gives their ligands, growth factors, and neurotransmitters the ability to act as signals that connect nerves and cancer cells. Axon guidance or the generation of new axons into a tumor and angiogenesis can both be guided by neurotrophins. This was demonstrated using the expression patterns of the Tyrosine receptor kinases (Trks), with different Trk receptors binding different neurotrophins: NGF binds to TrkA, BDNF binds to TrkB, and neurotrophin 3 (NT3) binds to TrkC. The expression of these Trks was established in tumors from oral squamous cell carcinoma (OSCC) patients as well as on the surface of high metastatic cells grown in culture. In both cases, the Trk receptors TrkB and TrkC were overexpressed. The expression levels of these receptors in OSCC patients correlated with a poor prognosis and lower survival rates [86]. The same was true for increased expression of TrkB in patients with ovarian cancer [87]. Some neurotrophies, such as NGF and BDNF, can promote angiogenesis independently of VEGF and may be the reason why some tumors are resistant to anti-VEGF therapy [88].

Neurotransmitters can also link neurogenesis, angiogenesis, and cancer. Increased levels of the dopamine receptor D2R are associated with lower survival rates in gastric cancer [89]. Dopamine is also able to stimulate the EGFR-AKT pathway, stimulating migration and invasion [24]. The increased expression of the D2R receptor has been observed in multiple cancers [90,91].

### 6.1. FGF2

Fibroblast growth factor (FGF-2) and its receptor FGFR2 are able to stimulate endothelial cell proliferation [92]. FGF2 is known to be overexpressed in dividing neural stem cells and its overexpression leads to an increased proliferation of neural stem cells [93]. It has also been shown that increased FGF-2 expression in a rat stroke model leads to increased neurogenesis and can lead to the recovery of brain function following a stroke [94]. Upregulation of IGF-1 is also observed in angiogenesis [92]. It is also responsible for regulating migration and ECM degradation during angiogenesis where it increases the expression of Urinary-Type Plasminogen Activator and matrix metalloproteinases in endothelial cells [95]. FGF2 stimulation of endothelial cells leads to the formation of membrane vesicles containing MMP-2, MMP-9, TIMP-1, and TIMP-2 on the cell surface. These vesicles assist in the formation of capillary-like structures [96]. Like many other growth factors, the activation of the Ras/ERK pathway by the binding of FGF3 to FGFR results in the expression of pro-angiogenic and pro-neurogenic genes (Figure 2).

### 6.2. IGF-1

Insulin-like growth factor-1 (IGF-1) is known to play a role in the development of the brain; despite this, it is normally expressed in the liver. IGF-1 is able to induce the proliferation of adult neural stem cells in the hippocampus. This proliferation occurs as a result of the activation of the MAPK [97]. Not only does IGF-1 have neurogenic/neurotrophic properties but it is also able to induce angiogenesis [98]. An experiment using a mouse model of permanent focal ischemia was used to illustrate the overexpression of IGF-1 using an adeno-associated viral (AAV) vector with the IGF-1 gene cloned into the vector. This overexpression resulted in higher levels of vascular density and increased neurogenesis noted the day after ischemic injury. This is accomplished through IGH-1 activating ERK, which leads to the generation of vascular endothelial cells and the initiation of PI3K and AKT signaling (Figure 3) [99]. IGF-1 also induces the activity of the Brain-4 (Brn4) transcription factor. This leads to the promotion of neural differentiation in neuronal stem cells in the hippocampus [49].

### 6.3. VEGF

Vascular endothelial growth factor induces angiogenesis after binding to its receptor on endothelial cells. The role played by VEGF in angiogenesis is well-understood but recent evidence suggests that it also plays a role in neuronal growth, survival (neurotrophic), axonal outgrowth (neurotropic), and neuroprotection. It has been established that VEGF has direct effects on neurons and glial cells, leading to increased growth (including axonal outgrowth) and survival (Figure 4). VEGF has also been implicated in multiple neurological disorders [100]. These direct effects are due to the presence of VEGF receptors on the surface of neuronal cells. These receptors include Flk-1 and Flt-1 [101].

### 6.4. BDNF

Brain-derived neurotrophic growth factor is known to induce neurogenesis. It has been shown that intrahippocampal administration of BDNF increased neurogenesis in an adult rat model. Intracerebroventricular infusion of BDNF led to increased production of new nerve cells in the olfactory bulb of adult rats [102]. Overexpression of BDNF using an AAV vector resulted in higher levels of neurogenesis, and in a mouse model of stroke injury resulted in better recovery following ischemic injury [103]. In mouse models of myocardial infarction, some mice were wild-type while others were BDNF (+/−) heterozygous. The heterozygous mice showed better recovery rates as well as reduced levels of vascularization. This indicates that BDNF is able to induce angiogenesis [104] (Figure 5). BDNF stimulates angiogenesis by recruiting bone-marrow-derived cells as endothelial progenitor cells. BDNF also stimulates stem cells to differentiate into endothelial cells [105].

### 6.5. NGF

The attraction of nerve fibers to tumor cells is facilitated by cancer cells secreting neurotrophic growth factors. [roNGF is expressed in prostate cancer cells, where, once it is processed to nerve growth factor, drives nerve infiltration. The levels of proNGF correlated with tumor aggressiveness, with low-risk tumors showing significantly lower levels [106]. Prostate cancer cells are also able to stimulate neuron outgrowth by releasing proNGF (Figure 6) [106]. There are some doubts as to whether proNGF needs to be processed into NGF to perform these functions, with some researchers believing that proNGF itself is responsible for these actions. Other processes regulated or stimulated by proNGF/NGFs include perineural invasion [7] and tumor neoneurogenesis [106].

The role played by NGF as a neurotrophin, promoting neurotrophic and neurotropic effects in sympathetic neurons, is well known. Recent evidence suggests that NGF also plays a role in angiogenesis. Angiogenesis involves changes and migration in endothelial cells (ECs). It was found that NGF was able to induce the migration of cultured endothelial cells to the same extent as VEGF [107]. Another study reported similar results using the chorioallantoic membrane (CAM) of a quail as a model system. Due to the highly vascularized nature of this membrane, quail is an ideal model to study angiogenesis. NGF was found to have pro-angiogenic effects on the natural vascularization of these membranes and, once again, this effect was similar to that induced by recombinant VEGF_165_a (rhVEGF) [108]. These effects of NGF on EC migration and CAM vascularization could be blocked by the addition of the NGF receptor antagonist K252a[(8R*,9S*,11S*)-(/)-9-hydroxy-9-methoxycarbonyl-8-methyl-2,3,9,10-tetrahydro-8,11-epoxy-1H,-8H,11H-2,7b,11a-triazadibenzo(a,g)cycloocta(c,d,e)trindene-1-one]. This selective blocker targets the NGF/trkA receptor [107,108]. The similar VEGF blocker SU-5416, which targets the VEGF/Flk1 receptor, had no effect on the proangiogenic effect of NGF. Therefore, NGF has a direct angiogenic effect, relying on a distinct signaling pathway from the pro-angiogenic effect induced by VEGF (Figure 6) [108]. However, the study on the effects of NGF on EC migration indicated that the signaling pathways activated by the growth factors are able to activate each other’s downstream tyrosine kinase signaling pathways [107]. NGF also promotes VEGF expression through the MAPK-ERK2 phosphorylation signaling pathway [109].

### 6.6. Nestin

Nestin is a class VI intermediate filament that is normally expressed in nervous tissue in mammalian embryos. It is also found in the adult brain where it is found in the subventricular zone. This zone is where neurogenesis occurs. Nestin plays a role in brain development where it promotes survival, renewal, and mitogen-stimulated proliferation of neural progenitor cells. It also plays a role in mitosis, where it disassembles phosphorylated vimentin intermediate filaments (IFs). Nestin expression is also upregulated in various types of tumors. In these tumors, nestin expression was found to occur in the vascular endothelial cells, the same areas where angiogenesis takes place [110].

### 6.7. Neuropilins

The neuropilins (NRPs) are plasma membrane spanning receptors that have no cytosolic protein kinase domain, meaning that they only function as co-receptors of other receptors for various ligands. They were originally identified as co-receptors for semaphorin (reviewed in [111]) and vascular endothelial growth factor [112]. By acting as a co-receptor for semaphorins, they play a role in axon guidance during axon and neurogenesis, while as co-receptors for VEGF, they play a role in angiogenesis. Commonly, this role in angiogenesis is related to providing new neurons with blood supply [113,114]. In addition, they also contribute to the regulation of signaling pathways, such as the STAT, RAS, MAPK, PI3K, Notch, TGF-β, Wnt/β-catenin, and hedgehog pathways. This allows them to play a role in processes, such as remyelination, immune response, angiogenesis, cell survival, migration, and invasion [115,116,117]. The expression of NRP is known to be altered in tumors [118], with more advanced-stage tumors showing higher levels of expression [119,120]. The overexpression of these NRPs leads to leaky hypervascularization [121].

There are two members of the family, Neuropilin-1 (NRP-1) and Neuropilin-2 (NRP-2) [122]. NRP1 is known to be vital for the vascularization of the spinal cord, hindbrain, and retina in mouse models [121,123,124]. This confirmed the need for NRP1 in the vascularization of the CNS. This is due to NRPs acting as an adhesion molecule and receptor for two particular classes of semaphoring and VEGF, namely class 3 semaphorin (SEMA3A) and VEGF_165_a [125]. Alternative splicing gives rise to both membrane-bound and soluble isoforms of NRPs. Different forms of NRP are also created by the inclusion of different lengths of amino acids to the C-terminal [126].

## 7. The Role of Reactive Oxygen Species

As they multiply, malignant cells require nutrients and oxygen to survive and continue growing (reviewed in [127]). Cancer cells that are deprived of oxygen respond to local hypoxic conditions by releasing signaling molecules, such as chemokines and growth factors, that alter the tumor microenvironment. These signaling molecules interact with immune, endothelial, and neuronal cells and induce their migration to the primary tumor [60,128]. Angiogenesis is also regulated via the crosstalk between reactive oxygen species (ROS) and Ca^2+^ ions. The ROS, nitric oxide, activates Ca^2+^ channels. This occurs via the glutathionylation of the Serine incorporator (SERINC) protein and the involvement of the NADPH oxidase (NOX) protein family. This process is also known as the VEGF-dependent anti-ROS angiogenic pathway as VEGF can induce this process. This is demonstrated by the induction of endothelial cell migration by H_2_O_2,_ which, in turn, can be inhibited by catalase and superoxide dismutase (SOD). The migration of these endothelial cells is also accompanied by an influx of Ca^2+^ into these cells [129]. This process is reliant on NOX2 or Nox4 activity. H_2_O_2_ supplementation can bypass the lack of NOX4 but not Nox2. This indicates that NOX2 is acting downstream of both H_2_O_2_ and Nox4. The production of ROS by Nox4, in turn, is activated by VEGF. Certain ROS can also activate the expression or stimulate the activity of transcription factors that regulate angiogenesis [130].

The expression of both NOX2 [131] and *Nox 4* [132] has been found to be expressed in neurons. Nox1 expression was also found to increase in response to nerve growth factor (NGF) signaling [133]. In neurons, NADPH oxidases play roles in the modulation of the activity of neurons as well as altering the cell fate or neurons. Brain-derived neurotrophic factor signals for neurons to undergo apoptosis as a result of serum deprivation, a process which is modulated by NOX2 [134]. NOX1 also negatively regulates neurite outgrowth [133].

Ets-1, NF-kB, and STAT-3 ROS also induces the expression of various genes involved in angiogenesis, such as monocyte chemoattractant protein-1 (MCP-1), vascular cell adhesion molecule 1 (VCAM-1), and matrix metalloproteinases (MMPs).

The contribution of the nervous system to cancer progression was demonstrated in mouse models of gastric cancer. Here, the surgical or pharmacological denervation of the stomach suppressed gastric tumorigenesis [135]. Denervation resulted in decreased Wnt signaling and suppressed stem cell activity. The decrease in stem cell expansion was found to be linked to cholinergic signaling. In the mouse stomach, cholinergic nerves regulate gastrointestinal epithelial proliferation [136]. Such denervation has been tested as a therapeutic strategy to treat gastric cancer where it can be used in conjunction with chemotherapy [135]. Nitric oxide (NO) is able to initiate angiogenesis through the generation of eNOS and hydrogen sulfide (H_2_S). H_2_S is generated via the action of cystathionine-γ-lyase (CSE) [137].

## 8. Therapeutic Targeting of the Interplay between Angiogenesis and Neurogenesis

Targeting angiogenesis for therapy in cancer is a well-established strategy that involves targeting the communication between tumor cells and the nearby blood vessels. It is known that interfering with neurogenic signaling can affect cancer development and progression. In mouse and tissue culture models of both prostate and lung cancer, it has been shown that chemical (6-hydroxydopamine, 6-OHDA) and surgical (hypogastric nerve cut) sympathectomy can prevent the development and progression of these cancers [5,73]. Since it appears that angiogenesis is triggered by axonogenesis and neurogenesis through adrenergic signaling, targeting the adrenergic receptor or ligands is a viable treatment option. For instance, perineural invasion (PNI) is a promising therapeutic target. PNI is induced by the activation of the β2-adrenergic receptor, leading to PKA/STAT3 activation, which, in turn, activates NGF, MMP2, and MMP9 expression. Ligands which bind to and activate this receptor in PNI include sympathetic fiber-derived noradrenaline. This can be achieved using drugs, such as propranolol and penbutolol, which are β Adrenergic blockers, or atropine and hyoscine, which are muscarinic antagonists. Studies have shown that these drugs prevent prostate cancer cell migration [138]. Dopamine receptors are also potential drug targets. In mouse models of lung cancer, Dopamine receptor D2 (D2R) agonist inhibits angiogenesis [91]. Dopamine (DA) inhibitors can prevent cancer cell proliferation. These inhibitors lead to the downregulation of ERK1/2 and PI3K/AKT pathways [139]. Both NGF and BDNF rely on Trk receptors to initiate the signaling pathways. The antagonism or inhibition of these Trks could potentially prevent neurotrophin signaling in cancer progression and initiation. This can be used to treat neuropathic pain related to cancer as well as inhibiting neurogenesis, angiogenesis, and their interplay. The Trk receptor inhibitors Larotrectinib and entrectinib have both been approved for use in the treatment of tumors [140]. It has long been known that innervation of gastric tissue promotes the development of gastric cancers. The severing of the vagus nerve at particular branches has been found to inhibit the development of gastric cancers in a mouse model. In addition to this, similar results were achieved when the mice were treated with BOTOX, which acted to block neural signaling [135].

## 9. Conclusions

The management and treatment of cancer require the advancement of knowledge concerning how tumors interact with their microenvironment. While angiogenesis has been known to contribute to the development and progression of cancer, it is now known that in some solid tumors, infiltrating nerves and catecholaminergic signaling may play an important role in tumor initiation and progression. Angiogenesis and neurogenesis share many of the same signaling pathways and both have a requirement for the rearrangement of tissue as well as the migration of cells. This provides the basis for the interplay between them. The two processes, angiogenesis and neurogenesis, are both activated by cancer cells and share many of the signaling pathways and molecules. This means that both processes are activated in a similar way and can activate each other. This also explains how cancer cells can take advantage of this dual activation to proliferate, migrate, and metastasize. This also makes this interplay an attractive target for the development of new therapies. These therapies can target multiple pathways while only targeting a few components that are shared between these processes (Figure 7). Although not covered in this review in detail, the interplay and signaling pathways involved in these processes also suppress the immune system. This may enable therapies targeting individual molecules to target three hallmarks of cancer. Much of our knowledge on the interplay between these processes comes from animal models of stroke and neurodegenerative disorders. It would be useful if future studies on the role and interplay of these processes in cancer are performed using cancer-specific models. Further knowledge of the interplay between neurogenesis, angiogenesis, nerve–cancer crosstalk, and the neuro–immune axis is required for the implementation of use of anti-neurogenic and anti-angiogenic targets in the treatment of cancer.

## Figures and Tables

**Figure 1 cancers-15-01805-f001:**
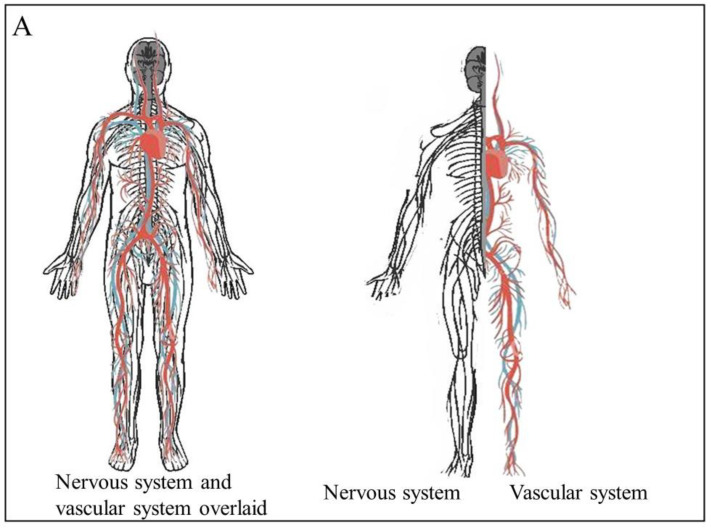
The physiological relationship between the vascular and nervous systems. (**A**) The vascular and neural systems mirror each other in the body as they both are required to service the entire body (**B**). The structure of the neurovascular unit in the brain and the blood brain barrier.

**Figure 2 cancers-15-01805-f002:**
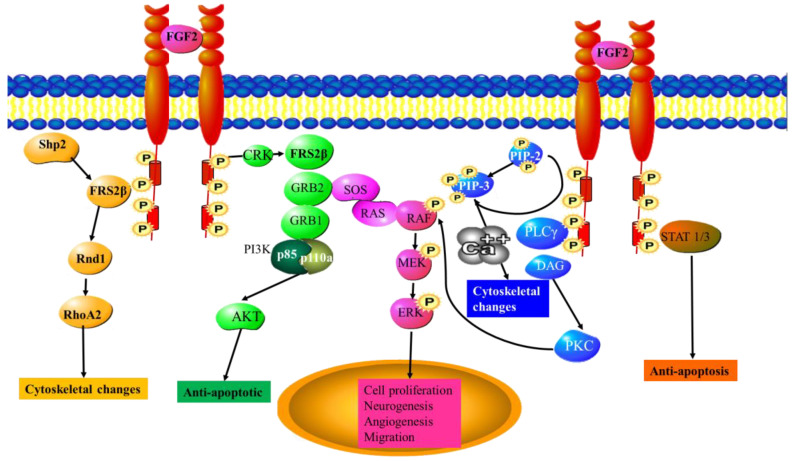
FGF-2 signaling pathways. The binding of FGF-2 to the FGF receptor FGF-R. The FGF-2 signaling pathway activates the Ras/ERK pathway, which leads to the activation of pro-angiogenic and pro-neurogenic genes. ERK—extracellular signal-regulated kinase, FRS2β—fibroblast growth factor (FGF) receptor substrate 2, GRB1/2—Growth factor receptor-bound protein 1/2, MEK—Mitogen-activated protein kinase, PI3K—Phosphoinositide 3-kinase, Rnd1—Rho-related GTP-binding protein, Shp2—Src homology region 2, SOS—Son of Sevenless.

**Figure 3 cancers-15-01805-f003:**
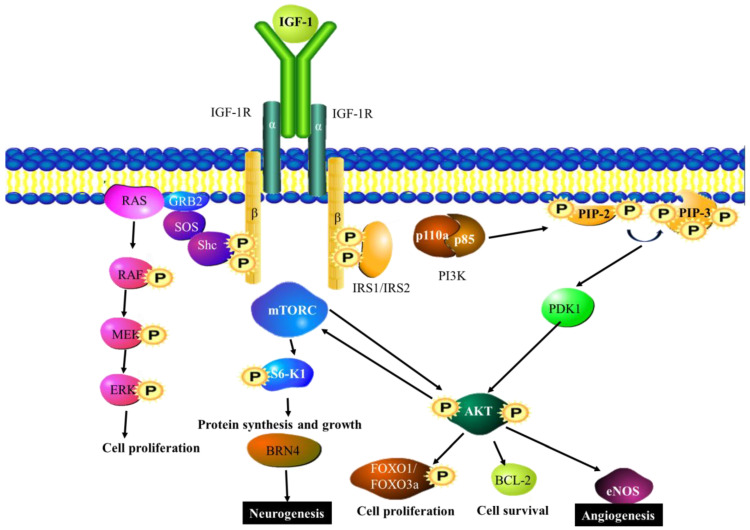
IGF-1 signaling pathways: The binding of IGF-1 to the IGF receptor FGF-1R. The IGF-1 signaling pathway activates the Ras/ERK pathway, which leads to the activation of pro-angiogenic and pro-neurogenic genes. In addition to this, the pathway activates the Brn-4 transcription factor, which stimulates neuronal growth and differentiation. BRN4—POU domain, class 3, transcription factor 4, eNOS—Endothelial NOS, ERK—extracellular signal-regulated kinase, FRS2β—fibroblast growth factor (FGF) receptor substrate 2, FOXO 1/3a—Forkhead family of transcription factors, GRB1/2—Growth factor receptor-bound protein 1/2, IRS1/2—Insulin receptor substrate 1/2, MEK—Mitogen activated kinase, mTORC—mammalian target of rapamycin complex 1, PDK1—Pyruvate dehydrogenase kinase isozyme 1, PIP-2/3—phospholipase, PI3K—Phosphoinositide 3-kinase, Shp2—Src homology region 2.

**Figure 4 cancers-15-01805-f004:**
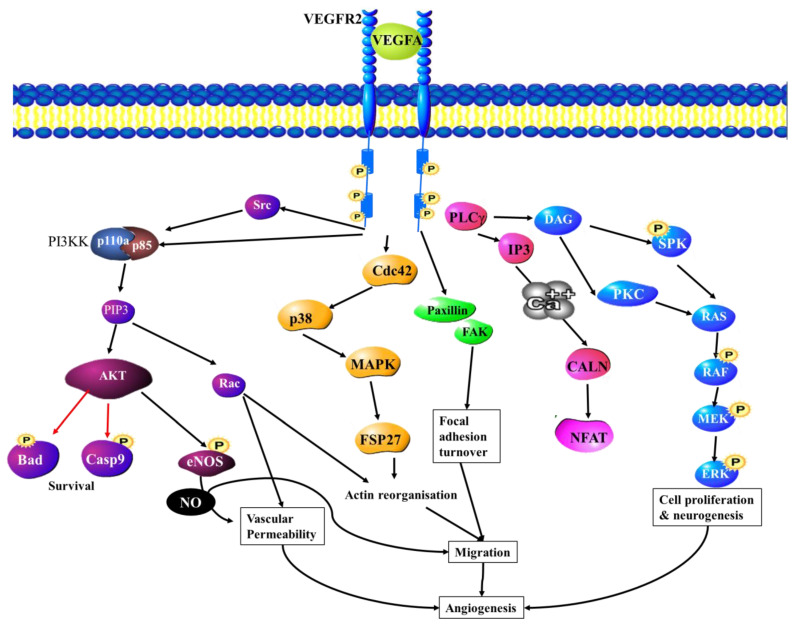
VEGF signaling: VEGFR-2 activation stimulates migration and extracellular matrix (ECM) invasion. The BEGF pathway results in neurogenesis, which leads, in combination with increased vascular permeability, to increased angiogenesis. CASP9—Caspase9, eNOS—Endothelial NOS, ERK- extracellular signal-regulated kinase, FAK—Protein-tyrosine kinase, FRS2β—fibroblast growth factor (FGF) receptor substrate 2, FOXO 1/3a—Forkhead family of transcription factors, FSP-27—Fat specific protein 27, IRS1/2—Insulin receptor substrate 1/2, MEK-Mitogen activated kinase, NFAT—Nuclear factor of activated T-cells, PIP-2/3-phospholipase, PI3K—Phosphoinositide 3-kinase, PKC—Protein kinase C, PLCγ—Phosphoinositide phospholipase C, Shp2—Src homology region 2, SPK-SR protein kinase.

**Figure 5 cancers-15-01805-f005:**
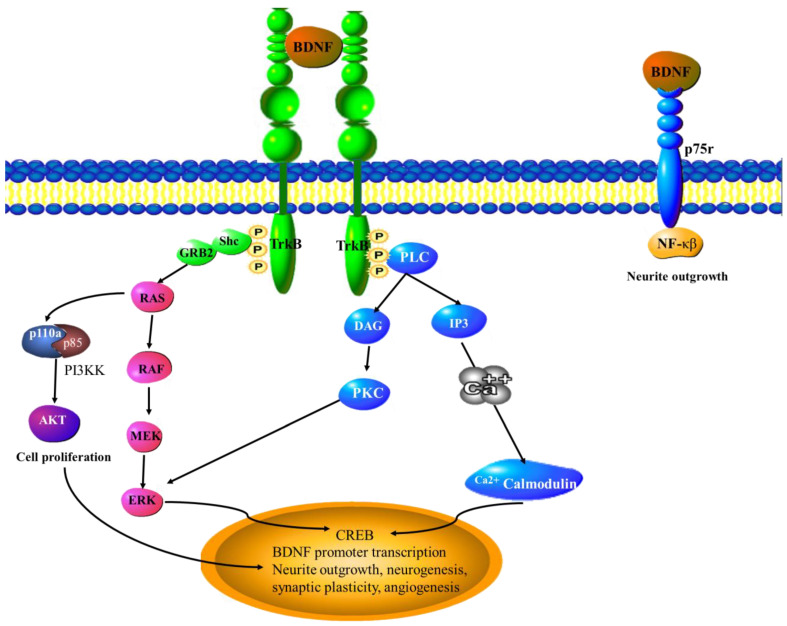
BDNF signaling pathways. Binding of BDNF to TrkB leads to the activation of MAPK, PI3K, and PLCγ pathways. The Shc adaptor protein recruits the growth-factor-receptor-bound protein 2 (grb2) to form a complex with Ras. This initiates ERK, which then activates the CREB transcription factor. Binding of TrkB also activates the PI3K and MF-κβ signaling pathways. These pathways play a role in neurotrophic functions (survival, growth, and differentiation). The PLCγ pathway generates inositol-1, 4, 5-triphosphate (IP3) and diacylglycerol (DAG), resulting in Ca^2+^/CaMKI activation and the initiation of the transcription of genes that are pro-angiogenic and pro-neurogenic. CREB—Cyclic AMP-responsive element-binding protein, ERK—extracellular signal-regulated kinase, GRB1/2—Growth factor receptor-bound protein 1/2, MEK—Mitogen activated kinase, PIP-2/3—phospholipase, PI3K—Phosphoinositide 3-kinase, PKC—Protein kinase C, PLCγ—Phosphoinositide phospholipase C, Shp2—Src homology region 2.

**Figure 6 cancers-15-01805-f006:**
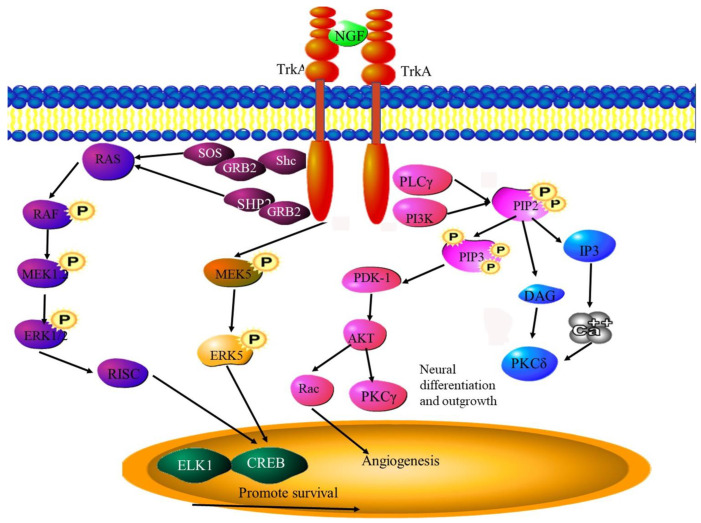
NGF signaling pathways. NGF binding to TrkA and p75NTR receptors promotes neuronal growth. The activation of the ERK signaling pathway promotes the transcription of pro-angiogenic and pro-neurogenic genes as in other neurotrophic and growth factor pathways. CREB—Cyclic AMP-responsive element-binding protein, ELK—ETS domain-containing protein, eNOS—Endothelial NOS, ERK—extracellular signal-regulated kinase, GRB1/2—Growth factor receptor-bound protein 1/2, MEK—Mitogen activated kinase, PDK—Pyruvate dehydrogenase kinase 1, PIP-2/3—phospholipase, PI3K—Phosphoinositide 3-kinase, PKC—Protein kinase C, PLCγ—Phosphoinositide phospholipase C, Shp2—Src homology region 2, SOS—Son of Sevenless.

**Figure 7 cancers-15-01805-f007:**
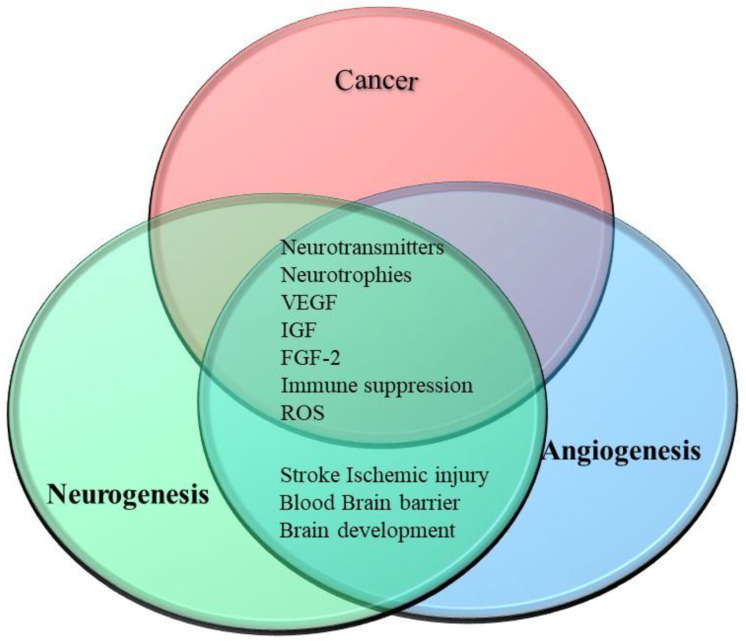
Relationship between neurogenesis and angiogenesis in cancer. The interplay between angiogenesis and neurogenesis is due to signaling molecules, such as neurotrophins, neurotransmitters, and growth factors, which all act to stimulate both processes. These processes can also be stimulated by reactive oxygen species. Both the angiogenic and neurogenic signaling pathways also act to suppress the immune system. Apart from cancer, the interplay between these two processes can best be observed in patients recovering from stroke or ischemic injury.

## Data Availability

Not applicable.

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
