# Peer review of "Genomic Interplay between Neoneurogenesis and Neoangiogenesis in Carcinogenesis: Therapeutic Interventions"

_cancers, 2023, doi:10.3390/cancers15061805_

Round 1
Reviewer 1 Report
General comments
This is an interesting review manuscript that summarizes the interplay between neurogenesis and angiogenesis and its effect on carcinogenesis. The authors have also discussed the potential therapeutic significance. However, some concerns remain.
Major concerns:
1. The authors did not discuss the role of the interplay between neogenesis and angiogenesis in the process of cancer initiation.
2. The discussion for therapeutic significance should be deeper. From a clinical view, the simplest way to block the effect of neurogenesis on carcinogenesis is to cut off neurons (neurotomy). Does anyone try it?
Author Response
Dear Reviewer,
Thank you for taking time to review the manuscript.
Please see attached.

Reviewer 2 Report
Present paper is very interesting and well structured. It contains most of the iportant informations in the field but Neuropilins family is completely missing. PLease insert in the manuscript a subheading including neuropilins dual role in angiogenesis and neurogenesis with focus on neuropilins and their relationship with endothelial markers of activated angiogenic process. Please insert a chapter about interrelation in between angiogenic and neurogenic factors in experimental models. References must be improved and updated.
Author Response
Dear Reviewer,
Thank you for taking time to review the manuscript.
Please see the attached

Reviewer 3 Report
1. It would be better if the authors revised lines no. 111-112.
2. There are two main types of angiogenesis- sprouting and intussusception. The authors detailed the sprouting angiogenesis mechanism - lines 91-95 of the manuscript. But after that they use only the term angiogenesis. If the authors mention which of the types they refer to (only sprouting angiogenesis or instussusceptive angiogenesis or both of them), the clarity and accuracy of the manuscript may increase. If they considered the intussusceptive type also, it may be useful to add some data to the text.
3. The authors used in the conclusions parts from the results (line 440- figure 6). A more concise style of these could be useful.
Author Response

(The authors gave the same response as above.)

Round 2
Reviewer 1 Report
Authors have addressed my concerns now